# Linear Regression Machine Learning Algorithms for Estimating Reference Evapotranspiration Using Limited Climate Data

Soo-Jin Kim [1], Seung-Jong Bae [1,*] and Min-Won Jang [2,*]

1   Institutes of Green Bio Science and Technology, Seoul National University, Pyeongchang-gun 25354, Korea
2   Department of Agricultural Engineering, Institute of Agriculture and Life Science,
    Gyeongsang National University, Jinju-si 52828, Korea
*   Correspondence: bsj5120@snu.ac.kr (S.-J.B.); mwjang@gnu.ac.kr (M.-W.J.); Tel.: +82-33-339-5811 (S.-J.B.);
    +82-55-772-1933 (M.-W.J.); Fax: +82-339-5830 (S.-J.B.); +82-55-772-1939 (M.-W.J.)

**Abstract:** A linear regression machine learning model to estimate the reference evapotranspiration based on temperature data for South Korea is developed in this study. FAO56 Penman–Monteith (FAO56 P–M) reference evapotranspiration calculated with meteorological data (1981–2021) obtained from sixty-two meteorological stations nationwide is used as the label. All study datasets provide daily, monthly, or annual values based on the average temperature, daily temperature difference, and extraterrestrial radiation. Multiple linear regression (MLR) and polynomial regression (PR) are applied as machine learning algorithms, and twelve models are tested using the training data. The results of the performance evaluation of the period from 2017 to 2021 show that the polynomial regression algorithm that learns the amount of extraterrestrial radiation achieves the best performance (the minimum root-mean-square errors of 0.72 mm/day, 11.3 mm/month, and 40.5 mm/year for daily, monthly, and annual scale, respectively). Compared to temperature-based empirical equations, such as Hargreaves, Blaney–Criddle, and Thornthwaite, the model trained using the polynomial regression algorithm achieves the highest coefficient of determination and lowest error with the reference evapotranspiration of the FAO56 Penman–Monteith equation when using all meteorological data. Thus, the proposed method is more effective than the empirical equations under the condition of insufficient meteorological data when estimating reference evapotranspiration.

**Keywords:** linear regression; machine learning; Penman–Monteith; polynomial regression; reference evapotranspiration

## 1. Introduction

Evapotranspiration is an important factor in the hydrological cycle, and it is defined as the sum of the total amount of evaporation released into the atmosphere as water vapor from the surface of water or soil and the amount of transpiration released into the atmosphere through leaf surfaces [1–3]. Evapotranspiration is considered as the basic data for the hydrological cycle and rainfall-runoff model verification and development, and for climate and meteorological prediction models. It needs to be considered for the management of agricultural water, which includes the planning and design of hydrological facilities and irrigation plans [4–8]. Reliable water resource management and planning can only be achieved by accurately calculating evapotranspiration because evapotranspiration is sensitive to changes in the growth stage or meteorological conditions during crop production [1,7,9].

The primary meteorological factors affecting evapotranspiration include radiation, air temperature, humidity, and wind speed. Considering the reference crop (alfalfa or grass), the maximum amount of evapotranspiration affected only by meteorological factors in a state where the soil is completely saturated with moisture, i.e., potential evapotranspiration, is referred to as reference evapotranspiration ($ET_0$), and the actual evapotranspiration

value for each crop period is determined by the product of $ET_0$ and crop coefficient for each crop-growth period. $ET_0$ is calculated using only meteorological data according to the Food and Agriculture Organization (FAO) guidelines, which is an accurate model for use in any region [6,10]. Following the Penman, Blaney–Criddle, and Modified Penman equations, at present, the FAO recommends the FAO56 Penman–Monteith (FAO56 P–M) equation as the standard; furthermore, it is used in legal plans, such as the long-term comprehensive plan for water resources and the rationalization plan for agricultural water use in South Korea.

Despite the accuracy of the evapotranspiration calculation, the FAO56 P–M equation is difficult to apply to areas with insufficient or missing meteorological data [8,11]. This model is difficult to apply realistically in areas with a low meteorological observation density or short observation history because the installation and maintenance of a meteorological station that collects meteorological data is expensive and laborious [12–15]. Given these limitations, the Blaney–Criddle, Hargreaves, Thornthwaite, and Priestly Taylor equations, which can help estimate reference evapotranspiration values with minimal climatic parameters, such as average and daily temperature differences, have been used, even though they are not as accurate as the Penman–Monteith model [6,16–19]. However, the problems associated with low accuracy, such as overestimation compared to that using FAO56 P–M, have been identified as a limitation [1,16,19]. Meanwhile, techniques used for calculating the amount of evapotranspiration on a spatial continuum using the vegetation index and thermal imaging are being developed given the development of remote sensing technologies, such as drones and satellite images. However, there are still limitations in the scope of application because of the discrepancies in the ground observation values, low temporal resolutions, and lack of verification in a space other than the ground observation point [20].

Machine learning technology is useful for predicting time-series data, such as evapotranspiration, without the prior knowledge of physical processes. Various studies have attempted to estimate $ET_0$ from a small amount of meteorological data. Bellido-Jiménez et al. [13] tested a machine learning technique to estimate $ET_0$ from the temperature data of southern Spain, which is a semi-arid region. Among the testing algorithms, such as multilayer perceptron (MLP), generalized regression neural network, extreme learning machine (ELM), support vector machine (SVM), random forest (RF), and XGBoost, MLP and ELM that only use the highest and lowest temperatures and solar radiation data demonstrated the best performance, with a root-mean-square error (RMSE) of at least 0.67 mm. Mehdizadeh et al. [14] applied neuro-fuzzy inference algorithms, such as the adaptive neuro-fuzzy inference system (ANFIS), ANFIS shuffled frog-leaping algorithm, and ANFIS invasive-weed optimization in Iran, which has a dry climate, to calculate the evapotranspiration using only temperature data and an RMSE minimum value of up to 0.83 mm/day. Dimitriadou and Nikolakopoulos [21] tested the utility of multiple linear regression (MLR) for estimating the reference evapotranspiration ($ET_0$) in the Peloponnese, Greece. Sixteen regression models were tested that combined seven factors, including sunshine hours (N), mean temperature (Tmean), solar radiation (Rs), net radiation (Rn), wind speed (u2), vapor pressure deficit (es–ea), and altitude (Z). As a result, the MLR5 model using N, Tmean, u2, es-ea, and Rn was evaluated as the best performance (RMSE 0.28 mm/day, Adjueted $R^2$ = 98.1%). For the semi-arid regions of Brazil with a lack of meteorological data, Reis et al. [22] calculated evapotranspiration more accurately than the Hargreaves equation using artificial neural networks, multiple linear regression (MLR), and ELM algorithms using only temperature data. Dou and Yang [23] applied ANN, SVM, ANFIS, and ELM to four different grasslands, forests, arable land, and wetlands in the United States, Germany, Belgium, and Sweden, respectively, where the performance based on the temperature, sunlight, relative humidity, and soil temperature data was the most accurate in the forests and the least accurate in the wetlands. Granata [24] applied machine learning (RF, M5P, Bagging) for estimating crop evapotranspiration in Florida, which is a humid subtropical region, and he reported the best performance using all climate factors required for using the FAO56 P–M equation, as well as a sufficient performance with

an RMSE of 0.35–0.40 mm/day using only three factors, such as average temperature, solar radiation, and relative humidity. Raza et al. [25] revealed that the SVM algorithm is the best model for estimating the reference evapotranspiration when all meteorological factors used in the FAO56 P–M equation are used, despite the different climatic zones. Ferreira et al. [26] tested the ANN, RF, XGBoost, and multivariate adaptive regression splines algorithms that calculated the $ET_0$ using only temperature and humidity data, and they attempted to interpolate the prediction results by linear regression again to overcome the disadvantages of the meteorological conditions in Brazil. Choi et al. [27] tested the neural network model and MLR algorithm using all meteorological factors necessary for calculating the FAO56 P–M equation for two stations as independent variables, which reported that the performance of the MLR model was not inferior as long as there was no derivation of negative values. These aforementioned studies suggest that the appropriate algorithm for estimating the $ET_0$ may differ based on the spatial and climatic characteristics of the study subjects and composition of training data owing to the characteristics of machine learning technology that reproduces empirical patterns.

Suitable methods with dependable accuracy are needed to estimate the $ET_0$ due to its importance in meteorological, hydrological, and agricultural studies [28]. Especially recently, there has been an increase in the research and development for implementing smart agricultural water management; however, the meteorological data and evapotranspiration observations that support it rely on the observation network of the Korea Meteorological Administration with a low observation density. Although it is desirable to measure meteorological data and evapotranspiration directly on the farmland subject to water management, it is unrealistic in terms of cost and management to install the facilities required to measure all meteorological items necessary for calculating the $ET_0$ on farmland pavements or in each neighborhood. There is no choice but to use distant point data through spatial interpolation or an area-weighted method. Even if all meteorological factors cannot be observed, calculating reliable $ET_0$ by only observing minimum factors, such as air temperature, will allow building the observation system required for smart water management.

This study aims to develop a technology that enables the accurate estimation of $ET_0$, even under conditions where it is difficult to estimate FAO56 P–M $ET_0$ due to sparse meteorological data. Therefore, $ET_0$ was estimated based on a machine learning algorithm using limited data (only temperature data), and its applicability was reviewed through a comparison with empirical equations, such as those of Hargreaves [8], Blaney–Criddle [29], and Thornthwaite [30]. The machine learning algorithm used simple and easy-to-understand linear regression algorithms [31].

## 2. Methods

This study examined whether ETo can be satisfactorily estimated using a linear regression machine learning algorithm without using all the meteorological factors necessary for calculating the FAO56 P–M equation. As shown in Figure 1, twelve linear regression models were tested and the performance of the models was compared. A machine learning model was implemented using Python library's Scikit-Learn, Pandas, Numpy, and Matplotlib. The regression model was used by changing the degree options to 1, 2, and 3 based on the MLR and PR models in the PolynomialFeatures function of the Scikit-Learn library. Furthermore, the SGDRegressor function of the Scikit-Learn library was used to apply the SGD method for minimizing the loss function. The maximum number of iterations was limited to 1000, and learning was performed at an adaptive learning rate; this was performed because the method using the adaptive learning rate had a greater learning efficiency than that when repeatedly using the learning rate directly designated by the user [19,31–33]. Each model was trained on a daily, monthly, and annual scale, respectively. The learning performance between twelve regression models was compared. In addition, the applicability of the proposed method was reviewed, compared with the $ET_0$ experience equation calculated using only temperature data.

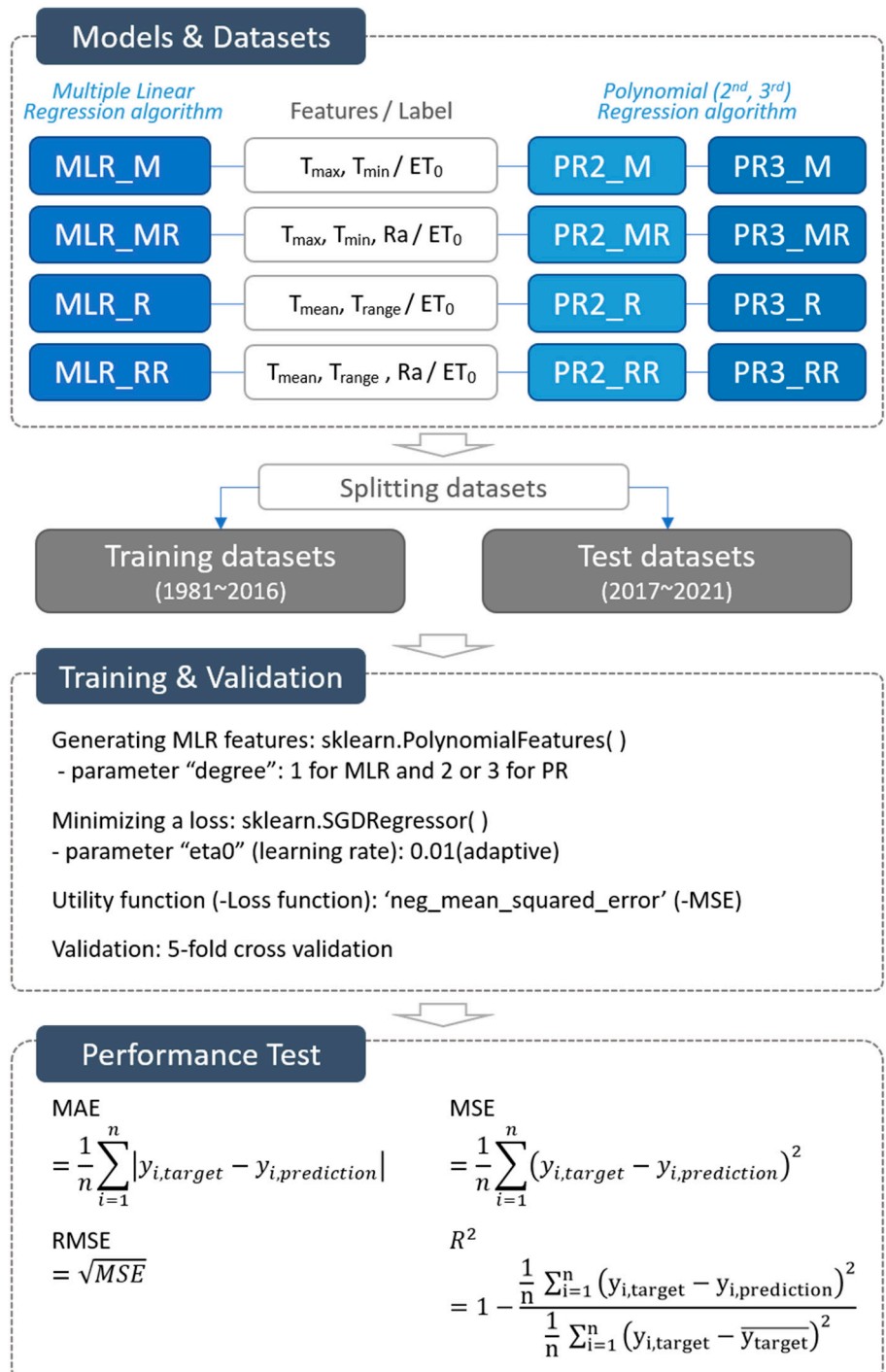

**Figure 1.** Procedure of $ET_0$ calculation using MLR and PR algorithms.

### 2.1. Linear Regression Machine Learning Models

The $ET_0$ calculation method was tested by focusing on the most basic linear regression analysis algorithm among machine learning algorithms. MLR and polynomial regression (PR) algorithms were applied based on the composition of independent variables. Linear regression learns relatively quickly, has a high explanatory power, and has no significant difference in performance compared to other algorithms [31,34–36].

Linear regression is a representative regression algorithm that models a linear relationship between a dependent variable and one or more independent variables. The

relationship between the independent and dependent variables is explained using the linear expression.

$$y = w_1x_1 + b \tag{1}$$

where y is the dependent variable (output value), $x_1$ is the independent variable (input value), $w_1$ is the weight for the independent variable, and b is the bias.

The MLR model is applied when one dependent variable and two or more independent variables are used as input data. This method is significant when each independent variable has a linear relationship with the dependent variable; however, the correlation between the independent variables is not high. The MLR model is expressed as

$$y = w_1x_1 + w_2x_2 + w_3x_3 + \cdots + w_nx_n + b \tag{2}$$

A PR model is a model that uses a linear model for learning nonlinear data; it is expressed as polynomials, such as quadratic and cubic equations, instead of the monomials of independent variables (Equation (2)). Quadratic and cubic polynomials were tested in this study.

$$y = w_2x_1^2 + w_1x_1 + b \tag{3}$$

In machine learning, linear regression is a learning model that identifies the parameter weights ($w_1$, $w_2$, ... , $w_n$) and deviation (b) that minimize the cost function (loss function) in a linear relationship. The cost function minimizes the loss by calculating the cost of the result predicted as an initial condition, i.e., repeating the process of calculating the loss, updating the model by modifying the parameters, and calculating the cost again, where the least-squares method to calculate the mean squared error (MSE) is often used. The least-squares method is a method of finding a parameter that minimizes the residual sum of squares between the target and predicted values, and is presented as follows:

$$MSE = \frac{1}{N} \sum_{i=1}^{N} \left( y_{i,target} - y_{i,prediction} \right)^2 \tag{4}$$

where $y_{i,target}$ is the i-th target value (actual value), $y_{i,prediction}$ is the i-th predicted value, and N is the number of data.

Gradient descent is applied to find the optimal solution by gradually moving the learning rate and to find the parameter that minimizes the loss function. Gradient descent is simply used in machine learning to obtain the values of a function's parameters that minimize a lose function as far as possible. In this study, stochastic gradient descent (SGD) was used among the gradient descent methods. SGD has received a considerable amount of attention just recently in the context of large-scale learning, and is suitable for regression problems with a high number of training samples (>10,000) [37,38]. Additionally, this method is more efficient than those using the entire batch, and it ensures more stable calculations than when using some randomized data [39].

Five-fold cross-validation was performed to use all independent variables for training and validation and to prevent overfitting or underfitting problems. The model was verified by dividing the training data into five equal parts, learning four parts (80%), and sequentially repeating the process of using the remaining one part (20%) as validation data five times.

The MSE, mean absolute error (MAE), RMSE, and coefficient of determination ($R^2$) were used as indices to evaluate the input data composition of the machine learning model necessary for calculating $ET_0$ and the performance according to the optimal model and for determining the best option.

$$MAE = \frac{1}{n} \sum_{i=1}^{n} \left| y_{i,target} - y_{i,prediction} \right| \tag{5}$$

$$MSE = \frac{1}{n} \sum_{i=1}^{n} \left( y_{i,target} - y_{i,prediction} \right)^2 \tag{6}$$

$$RMSE = \sqrt{MSE} \tag{7}$$

$$R^2 = \frac{SSE}{SST} = 1 - \frac{SSR}{SST} = 1 - \frac{\frac{1}{n} \sum_{i=1}^{n} \left( y_{i,target} - y_{i,prediction} \right)^2}{\frac{1}{n} \sum_{i=1}^{n} \left( y_{i,target} - \overline{y_{target}} \right)^2} = 1 - \frac{MSE}{var\left( y_{target} \right)} \tag{8}$$

where SSE is the explained variation sum of squares, SSR is the unexplained variation sum of squares, SST is the total variation sum of squares (SSE + SSR), and n denotes the number of test data.

The coefficient of determination ($R^2$) can be interpreted as the ratio of the variance of the predictable dependent variable in the independent variable. The coefficient of determination can take values between $-\infty$ and 1 according to the mutual relation between the ground truth and prediction model [40]. The assumption that the $R^2$ is greater than or equal to zero is based on the assumption that obtaining a negative $R^2$ discards the regression calculation in use and utilizes the mean value. That is, when $R^2$ is negative, it means an abnormal case in which the prediction performance is lower than the model predicted by the mean value. In particular, it can be negative when calculated on out-of-sample data (or for a linear regression without an intercept) [41].

### 2.2. Data Collection and Processing

In this study, daily data obtained from sixty-two meteorological stations were collected (Figure 2). The fifty-seven meteorological stations (excluding Jeju) had automated synoptic observation system (ASOS) data collected for more than 30 years, from 1981 to 2021. The five meteorological stations (Andong, Changwon, Taebaek, Bonghwa, and Cheorwon) collected data from 1983 to 2021, when continuous observation data existed.

The meteorological data between April and October (i.e., during the rice cultivation period) were limited to daily maximum temperature (DXT), daily minimum temperature (DNT), daily mean temperature (DMT), daily mean wind speed (DWS), daily relative humidity (DRH), daily solar radiation (DSR), or daily sunshine duration (DSD). From these collected data, 536,926 data were used as the research data by applying the average value of data collected from nearby observation stations. The daily $ET_0$ was calculated using the FAO56 P–M equation as the label data for machine learning (Table 1). Based on the FAO56 P–M equation, the national average annual $ET_0$ was 707.9 mm and the standard deviation was 59.6 mm. Figure 3 presents the spatiotemporal distribution of the average annual $ET_0$ from April to October. $ET_0$ was 711.1 mm/year from 1981 to 1990, 704.9 mm/year from 1991 to 2000, 706.3 mm/year from 2001 to 2010, and 720.0 mm/year from 2011 to 2021. In the past, $ET_0$ was high in the southern region, but recently, it has been on the rise in the central and northern regions of Gyeonggi-do.

### 2.3. Learning Model and Training Dataset

The learning model was constructed by varying the conditions of the available meteorological items to search for the daily $ET_0$ calculation model based on limited meteorological items. The daily $ET_0$ based on the FAO56 P–M equation was set as a label, and 12 input variable datasets were created: 4 for each learning algorithm, with different combinations of temperature and radiation items (Table 2). Among the input variables, the diurnal temperature range (DTR) and daily extraterrestrial radiation (DER) were, respectively, calculated and processed from the daily maximum and minimum temperatures and the latitude and longitude coordinates of the meteorological station. Extraterrestrial radiation was included in the training data to consider the effect of solar radiation while minimizing the omission of training data because solar radiation was not observed at all the meteorological stations.

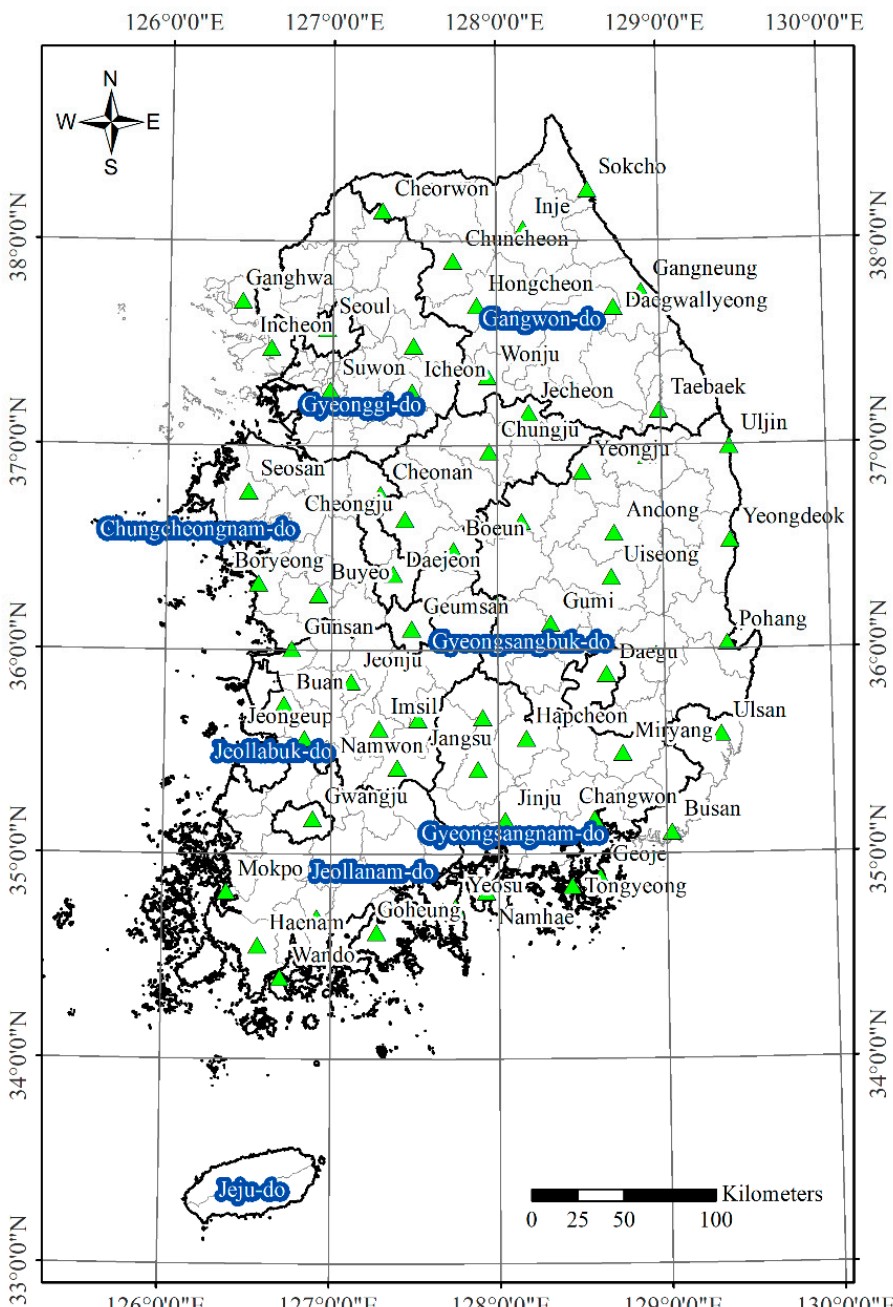

**Figure 2.** Distribution of sixty-two meteorological stations in South Korea.

**Table 1.** Nationwide average statistics of the collected climate data and calculated $ET_0$ for the irrigation period from April to October.

| Statistics | Daily Climate Data | | | | | | $ET_0$ by FAO56 P–M (mm/Year) |
|---|---|---|---|---|---|---|---|
| | DXT (°C/Day) | DNT (°C/Day) | DRH (%/Day) | DWS (m/s) | DSR (MJ/m²/Day) | DSD (h/Day) | |
| Maximum | 41.0 | 30.9 | 100.0 | 17.9 | 15.7 | 14.3 | 969.0 |
| Minimum | −1.5 | −12.4 | 12.1 | 0.0 | 1.5 | 0.0 | 423.7 |
| Average | 24.8 | 14.5 | 72.3 | 1.9 | 7.7 | 6.4 | 707.9 |
| Standard deviation | 5.4 | 6.6 | 13.4 | 1.2 | 2.0 | 4.1 | 59.6 |

DXT: daily maximum temperature, DNT: daily minimum temperature, DRH: daily relative humidity, DWS: daily mean wind speed, DSR: daily solar radiation, DSD: daily sunshine duration, FAO56 P–M: FAO56 Penman–Monteith.

For each learning model, a total of 470,586 data from 1981 to 2016 collected from 62 meteorological stations nationwide were used as training data; the test performance was evaluated with a total of 66,340 data obtained from 2017 to 2021. For testing the learning model for monthly and annual $ET_0$ calculations, a training dataset was added by processing the monthly and annual average values of the features in Table 2 for each observation station. In the monthly based training, a total of 17,563 training data and 2170 test data were used for 62 meteorological stations nationwide, whereas 2509 and 310 training and test data were used in the yearly based training.

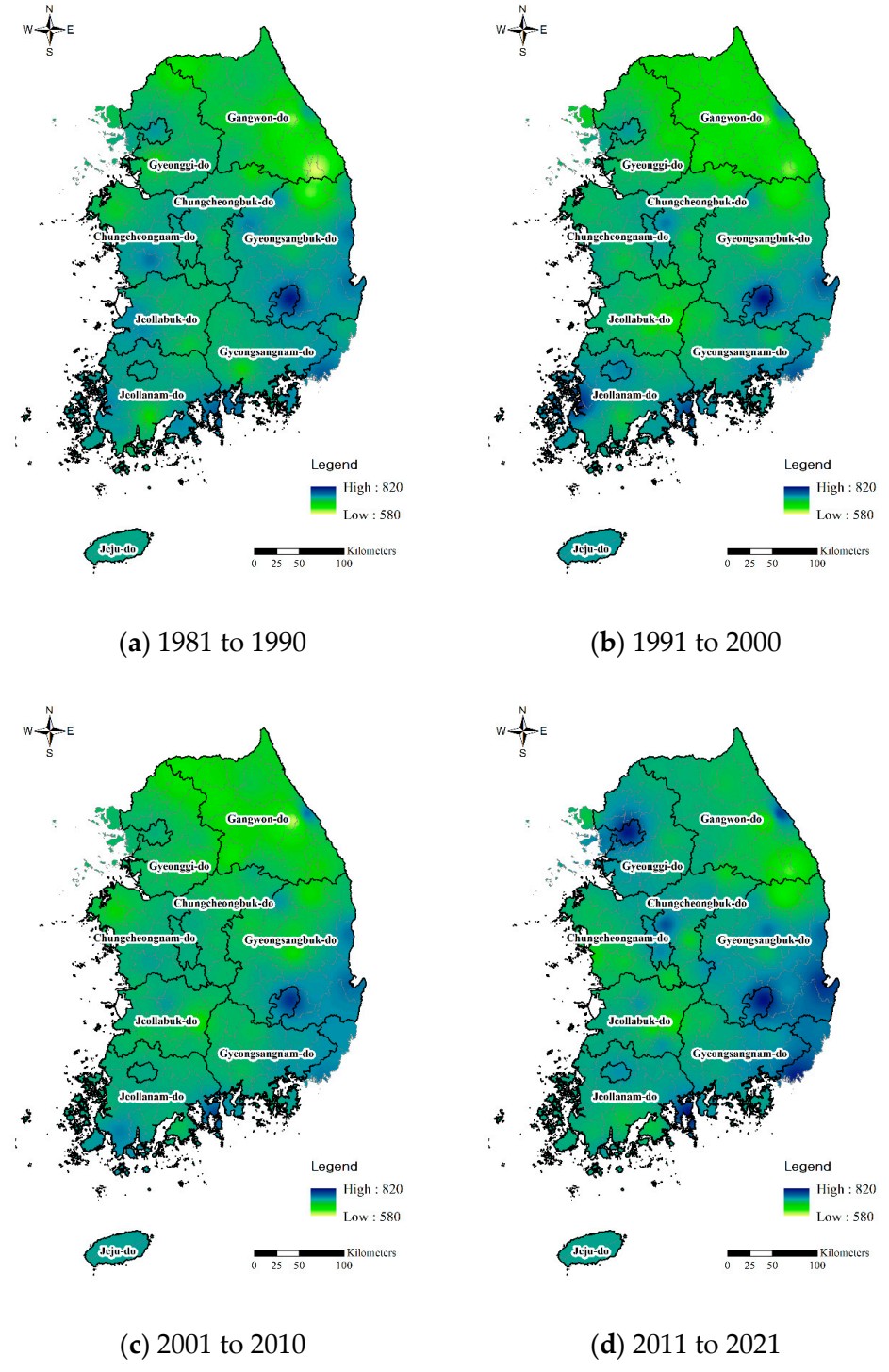

**Figure 3.** Spatiotemporal distribution of annual ETo.

**Table 2.** Twelve models of daily-based machine learning datasets.

| Models | Algorithms | Label | Features |
|--------|-----------|-------|----------|
| MLR_M | | | DXT, DNT |
| MLR_MR | Multiple linear | Daily ET$_0$ by FAO56 | DXT, DNT, DER |
| MLR_R | regression (MLR) | P–M | DMT, DTR |
| MLR_RR | | | DMT, DTR, DER |
| PR2_M | | | DXT, DNT |
| PR2_MR | Quadratic polynomial | Daily ET$_0$ by FAO56 | DXT, DNT, DER |
| PR2_R | regression (PR) | P–M | DMT, DTR |
| PR2_RR | | | DMT, DTR, DER |
| PR3_M | | | DXT, DNT |
| PR3_MR | Cubic polynomial | Daily ET$_0$ by FAO56 | DXT, DNT, DER |
| PR3_R | regression (PR) | P–M | DMT, DTR |
| PR3_RR | | | DMT, DTR, DER |

DXT: daily maximum temperature, DNT: daily minimum temperature, DER: daily extraterrestrial radiation, DMT: daily mean temperature, DTR: daily diurnal temperature range, and FAO56 P–M: FAO56 Penman–Monteith.

## 3. Results and Discussion

### 3.1. Performance of Models in Daily ET$_0$

The individual learning performances of the 12 models are presented in Table 3. The comparison of these performances indicates that the PR3_RR model trained as a cubic polynomial regression model using DMT, DTR, and DSR achieved the best results, with an MSE of 0.473 mm$^2$/day$^2$ (Table 3). For the test performance, all four performance indicators achieved the best results compared to the other models. Among the models that used only temperature data, the PR3_R and PR3_M models achieved the least errors, with RMSE = 0.888 mm/day and MAE = 0.724 mm/day. The PR algorithm showed an RMSE = 0.89 mm/day for the model that only learned temperature data without using extraterrestrial radiation data; this RMSE was less than that (RMSE = 0.94 mm/day) of the MLR algorithm. The model that learned extraterrestrial radiation together instead of using only temperature achieved a high performance in both training and test data regardless of the algorithm. For the R$^2$ obtained with the daily ET$_0$ using FAO56 P–M, all models learning DER showed a high explanatory power of 0.6 or more (Figure 4). Therefore, the method of using temperature and extraterrestrial radiation data together, and learning with the PR model instead of the MLR model, helped predict the daily ET$_0$ more accurately than that when using only the temperature.

**Table 3.** Performance of the applied models in training and test phases for daily ET$_0$ predictions.

| Models | Training | | Test | | | |
|--------|----------|----------|------|------|-----|-----|
| | MSE | 5-Fold MSE | MSE | RMSE | MAE | R$^2$ |
| MLR_M | 0.802 | 0.809 | 0.876 | 0.936 | 0.775 | 0.485 |
| MLR_MR | 0.586 | 0.595 | 0.642 | 0.801 | 0.643 | 0.623 |
| MLR_R | 0.801 | 0.809 | 0.874 | 0.935 | 0.773 | 0.486 |
| MLR_RR | 0.586 | 0.595 | 0.641 | 0.800 | 0.642 | 0.623 |
| PR2_M | 0.738 | 0.746 | 0.796 | 0.892 | 0.728 | 0.532 |
| PR2_MR | 0.495 | 0.504 | 0.547 | 0.740 | 0.579 | 0.678 |
| PR2_R | 0.733 | 0.741 | 0.793 | 0.891 | 0.728 | 0.534 |
| PR2_RR | 0.477 | 0.485 | 0.528 | 0.727 | 0.561 | 0.690 |
| PR3_M | 0.732 | 0.740 | 0.789 | 0.888 | 0.724 | 0.536 |
| PR3_MR | 0.480 | 0.488 | 0.529 | 0.727 | 0.564 | 0.689 |
| PR3_R | 0.730 | 0.738 | 0.788 | 0.888 | 0.724 | 0.537 |
| **PR3_RR** | **0.473** | **0.482** | **0.521** | **0.722** | **0.557** | **0.694** |

MSE: mean squared error, 5-fold MSE: Average value of MSE with K-fold cross-validation, RMSE: root-mean-square error, MAE: mean absolute error, and R$^2$: coefficient of determination. Bold font: best results.

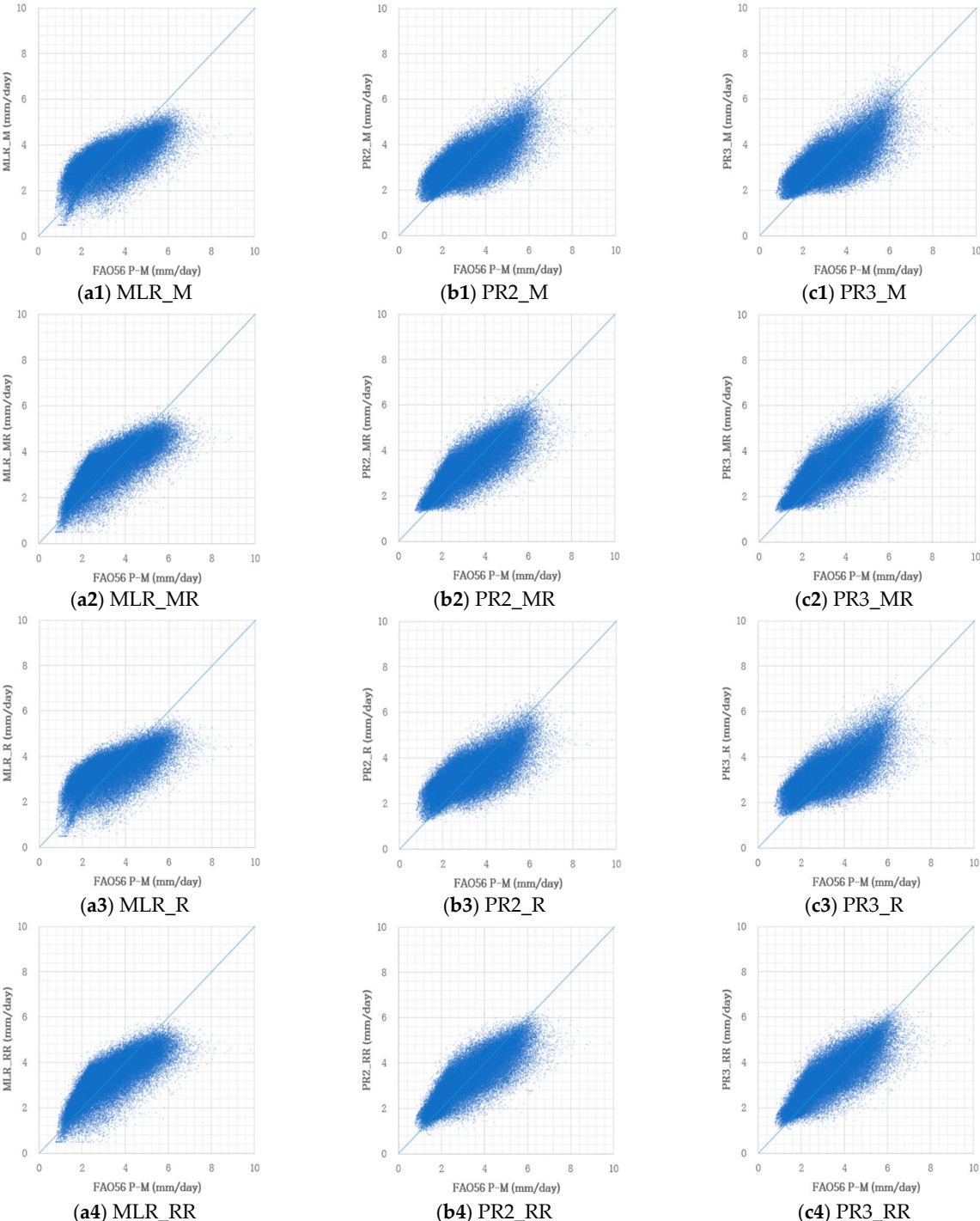

**Figure 4.** Scatter plots of the daily $ET_0$ calculated by FAO56 P–M and predicted by MLR and PR algorithms.

The error and regression fit of the monthly accumulated $ET_0$ from 2017 to 2021 using the learning model were analyzed against the monthly $ET_0$ (mm/month) calculated using FAO56 P–M for each national meteorological station (Figure 5). The PR3_RR and PR2_RR models achieved the least errors compared to the other ten models, and presented higher explanatory powers than the daily-based comparison with $R^2 = 0.78$. As indicated by the daily-based comparison, models using extraterrestrial radiation data as learning data presented less errors than models only using temperature data ($R^2$ was also higher than

0.75). The application of the PR algorithm was observed to be more appropriate than the MLR algorithm.

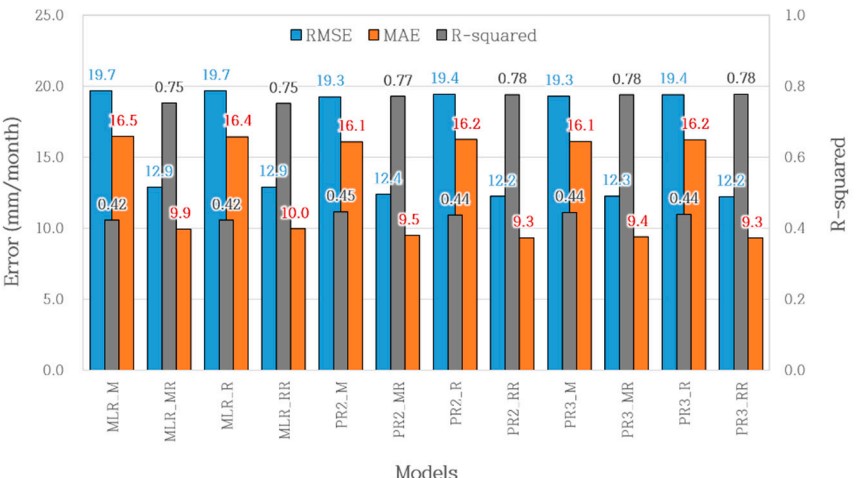

**Figure 5.** Comparison of monthly accumulated $ET_0$ using different models.

The comparison of the annual accumulated $ET_0$ from April to October with the annual $ET_0$ (mm/month) calculated using FAO56 P–M for each meteorological station in South Korea, as presented in Figure 6, indicated that the MLR_M model trained with only DXT and DNT with the MLR algorithm showed the best performance with an RMSE of 66.6 mm/year and an MAE of 51.5 mm/year. Unlike that of both the daily and monthly accumulated $ET_0$, the annual accumulated $ET_0$ presented the smallest error when using the learning result of an MLR algorithm. However, in all models, there was no correlation between the annual $ET_0$ obtained using FAO56 P–M and an $R^2$ of 0 or less. When $R^2$ was less than 0, it indicates an abnormal case whose prediction performance was lower than the model predicted by the average value. Therefore, it was not appropriate to calculate the annual $ET_0$ using daily learning results.

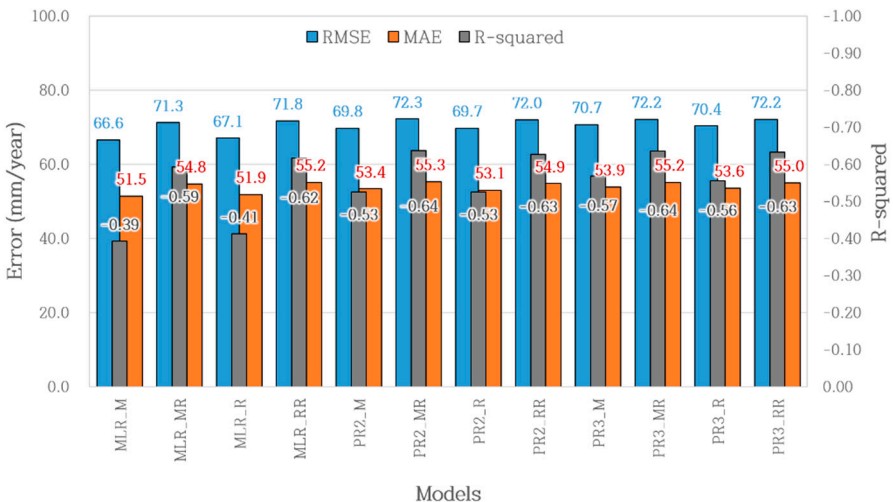

**Figure 6.** Comparison of annual accumulated $ET_0$ using different models.

### 3.2. Performance of Models in Monthly $ET_0$

The monthly average data of the independent variables for all twelve learning models presented in Table 3 were prepared to learn about monthly evapotranspiration for testing the application of linear regression machine learning technology for estimating monthly $ET_0$. The monthly sum of the daily $ET_0$ values calculated by FAO56 P–M was used as

the label, and the monthly average of DXT (MDXT), monthly average of DNT (MDNT), monthly average of DTR (MDTR), and monthly average of DER (MDER) were prepared as the training data.

Although the PR3_RR model was the best in daily learning, in monthly learning, the PR3_MR model presented a slightly better performance than PR3_RR in all phases of training, validation, and testing (Table 4 and Figure 7). The test performance of the PR3_MR model was evaluated as MSE = 127.69 mm$^2$/month$^2$, RMSE = 11.30 mm/month, MAE = 8.75 mm/month, and R$^2$ = 0.81. For the model that only learned temperature data by excluding extraterrestrial radiation, PR3_R using MDMT and MDTR as input data achieved the best results, with MSE = 19.28 mm$^2$/month$^2$ and MAE = 16.47 mm/month. However, there was a considerable difference in the performances from the models learning MDER together. Similar to daily learning, the performance was better when using PR rather than MLR, and when using extraterrestrial radiation data simultaneously with temperature data rather than only temperature data. Furthermore, R$^2$ presented greater explanatory power for monthly learning than daily learning. Instead of accumulating the daily ET$_0$, as presented in Figure 5, the approach of predicting the monthly ET$_0$ was better in terms of error and coefficient of determination.

**Table 4.** Performance of applied models in training and test phases for monthly ET$_0$ predictions.

| Models | Training | | Test | | | |
|---|---|---|---|---|---|---|
| | MSE | 5-Fold MSE | MSE | RMSE | MAE | R$^2$ |
| MLR_M | 333.21 | 337.51 | 401.31 | 20.03 | 17.12 | 0.40 |
| MLR_MR | 127.40 | 131.67 | 154.07 | 12.41 | 9.60 | 0.77 |
| MLR_R | 333.20 | 337.59 | 400.58 | 20.01 | 17.10 | 0.40 |
| MLR_RR | 127.40 | 131.77 | 153.77 | 12.40 | 9.59 | 0.77 |
| PR2_M | 321.98 | 326.43 | 384.69 | 19.61 | 16.70 | 0.43 |
| PR2_MR | 109.06 | 113.47 | 135.84 | 11.66 | 9.03 | 0.80 |
| PR2_R | 320.28 | 325.01 | 379.76 | 19.49 | 16.64 | 0.43 |
| PR2_RR | 109.04 | 113.76 | 136.26 | 11.67 | 9.05 | 0.80 |
| PR3_M | 320.76 | 325.21 | 381.89 | 19.54 | 16.65 | 0.43 |
| **PR3_MR** | **100.77** | **104.94** | **127.69** | **11.30** | **8.75** | **0.81** |
| PR3_R | 317.44 | 321.85 | 371.75 | 19.28 | 16.47 | 0.45 |
| PR3_RR | 101.84 | 106.45 | 128.84 | 11.35 | 8.81 | 0.81 |

MSE: mean squared error, 5-fold MSE: average value of MSE with K-fold cross-validation, RMSE: root-mean-square error, MAE: mean absolute error, and R$^2$: coefficient of determination. Bold font: best results.

*3.3. Performance of Models in Annual ET$_0$*

The learning model for estimating the annual ET$_0$ was tested using the accumulated ET$_0$ for 214 days from April 1 to October 31 calculated using FAO56 P–M as the label. From the daily meteorological data, the annual average of DXT (ADXT), annual average of the DNT (ADNT), annual average of the DTR (ADTR), and annual average of the DER (ADER) were prepared as the input data.

As with monthly learning, the PR3_MR model performed the best (Table 5). The test performance of the PR3_MR model was evaluated as MSE = 1639.7 mm$^2$/year$^2$, RMSE = 40.49 mm/year, MAE = 32.26 mm/year, and R$^2$ = 0.49. Among the models that only learned temperature data, PR3_M using ADXT and ADNT as input data achieved the best results, with RMSE = 41.32 mm/year and MAE = 32.94 mm/year. Similar to daily and monthly learning, the performance was better when using PR rather than MLR, and when using extraterrestrial radiation data simultaneously with temperature data rather than only temperature data. However, R$^2$ = 0.5 or less in all models, and the explanatory power was lower than that of the daily or monthly learning. Furthermore, compared with the results presented in Figure 6, the value predicted by learning the annual meteorological data was more appropriate than the value obtained by accumulating daily ET$_0$ estimates.

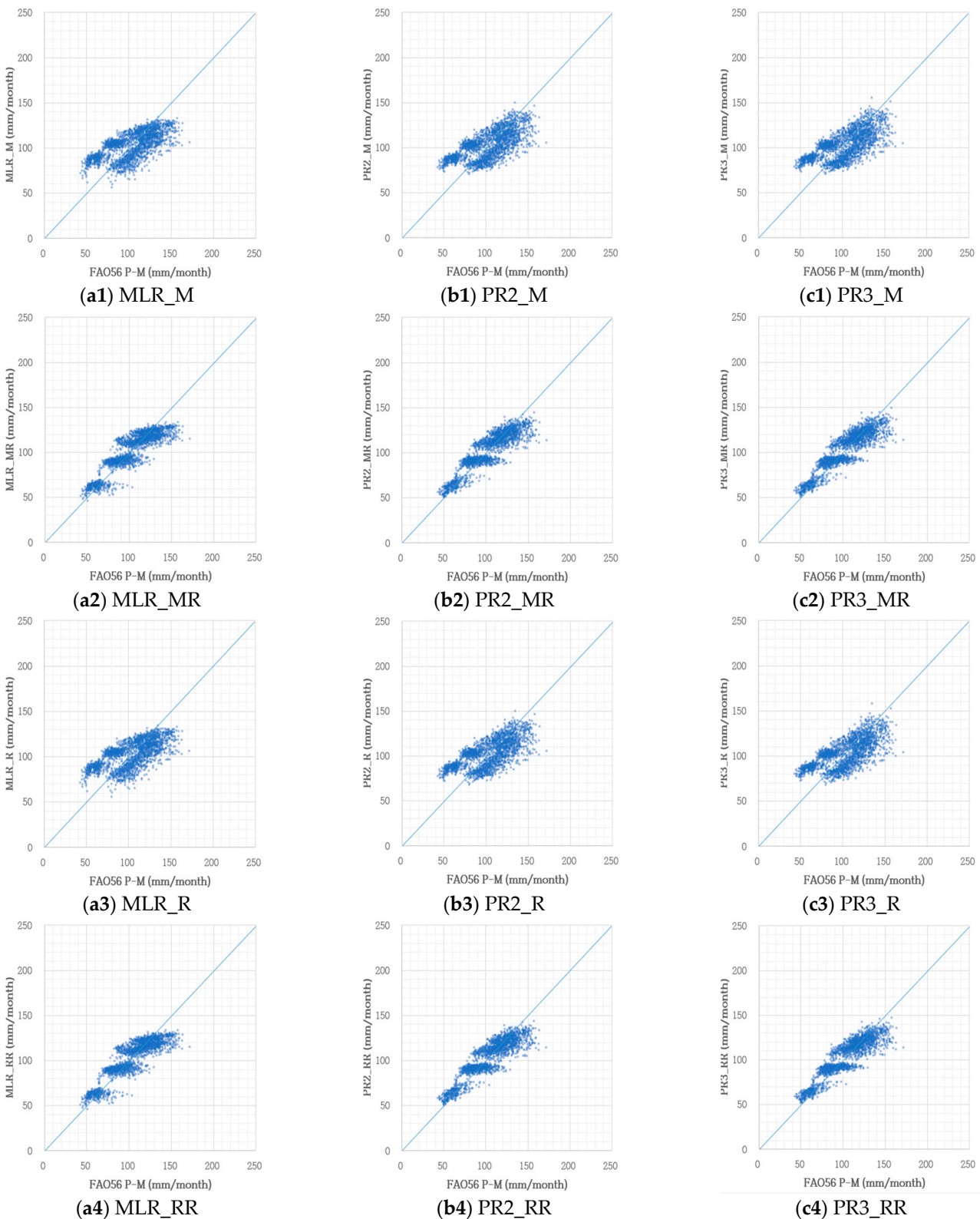

**Figure 7.** Scatter plots of monthly ET$_0$ calculated by FAO56 P–M and predicted by MLR and PR algorithms.

**Table 5.** Performance of the applied models in training and test phases for annual $ET_0$ predictions.

| Models | Training | | Test | | | |
|---|---|---|---|---|---|---|
| | MSE | 5-Fold MSE | MSE | RMSE | MAE | $R^2$ |
| MLR_M | 1837.7 | 1907.5 | 1921.0 | 43.83 | 34.49 | 0.40 |
| MLR_MR | 1836.4 | 1918.9 | 1924.0 | 43.86 | 34.51 | 0.40 |
| MLR_R | 1838.0 | 1907.1 | 1926.1 | 43.89 | 34.53 | 0.40 |
| MLR_RR | 1836.7 | 1918.6 | 1931.1 | 43.94 | 34.57 | 0.39 |
| PR2_M | 1792.4 | 1866.0 | 1787.5 | 42.28 | 33.65 | 0.44 |
| PR2_MR | 1776.0 | 1888.0 | 1720.5 | 41.48 | 33.07 | 0.46 |
| PR2_R | 1786.7 | 1866.2 | 1758.3 | 41.93 | 33.31 | 0.45 |
| PR2_RR | 1769.1 | 1887.7 | 1697.0 | 41.20 | 32.73 | 0.47 |
| PR3_M | 1780.3 | **1853.4** | 1707.4 | 41.32 | 32.94 | 0.46 |
| **PR3_MR** | **1761.1** | 1887.2 | **1639.7** | **40.49** | **32.26** | **0.49** |
| PR3_R | 1781.6 | 1860.9 | 1727.4 | 41.56 | 33.16 | 0.46 |
| PR3_RR | 1761.8 | 1892.6 | 1663.9 | 40.79 | 32.45 | 0.48 |

MSE: mean squared error, 5-fold MSE: average value of MSE with K-fold cross-validation, RMSE: root-mean-square error, MAE: mean absolute error, and $R^2$: coefficient of determination. Bold font: best results.

*3.4. Comparison with Temperature-Based Empirical Methods*

Methods, such as Hargreaves, Blaney–Criddle, and Thornthwaite, which can estimate $ET_0$ using temperature data alone, are used when there are insufficient observed meteorological data or when the reliability of the data is insufficient. These methods are recommended when calculating $ET_0$ for at least ten days, and the accuracy of calculating $ET_0$ decreases depending on the climate. However, these methods are still widely used in situations with limited meteorological data because they only require temperature data for the calculations [8,29]. Monthly and annual $ET_0$ were calculated using the Hargreaves, Blaney–Criddle, and Thornthwaite equations using only temperature data obtained from 2017 to 2021 for sixty-two meteorological stations nationwide to analyze the usefulness of linear regression machine learning technology. For the Hargreaves equation, which can also calculate daily evapotranspiration, the daily $ET_0$ was calculated and compared with the result obtained by machine learning [42].

The results of the PR3_RR and PR3_R models were compared with the Hargreaves equation's daily evapotranspiration calculation results (Table 6). Daily $ET_0$ according to FAO56 P–M used as the reference had maximum and minimum values of 9.80 and 0.76 mm/day, respectively; the range of the daily $ET_0$ by Hargreaves and the PR3_R and PR3_RR learning models was narrower. PR3_RR had the maximum and minimum values of 6.73 and 1.05 mm/day, respectively, in the narrowest range. The average of the daily $ET_0$ using the PR3_R and PR3_RR learning models was 3.37 mm/day, which was similar to the 3.38 mm/day of the FAO56 P–M. However, as shown in Figure 8, Hargreaves achieved a tendency of overestimation with 3.93 mm/day, compared to that of FAO56 P–M and PR3_RR. Comparing the error with FAO56 P–M, the Hargreaves equation presented a greater error with RMSE = 0.95 mm/day compared to PR3_R (0.89 mm/day) and PR3_RR (0.72 mm/day). Therefore, the PR learning model achieved an appropriate level of accuracy as an alternative to the Hargreaves equation when estimating the daily $ET_0$.

The results of PR3_MR and PR3_R and those of the Hargreaves, Blaney–Criddle, and Thornthwaite equations were compared with the monthly $ET_0$ by FAO56 P–M (Table 6). The national average of monthly evapotranspiration by FAO56 P–M was about 103.3 mm/month; the maximum value was 171.55 mm/month, and the minimum value was 42.25 mm/month. The averages of the Hargreaves, Blaney–Criddle, and Thornthwaite equations were 120.27 mm/month, 157.03 mm/month, and 105.76 mm/month, respectively. Figure 9 shows that the temperature-based empirical equation overestimates the monthly $ET_0$ compared to the FAO56 P–M. Such results are attributed to the limitations of the empirical equations, as reported in the previous studies conducted by [1,43,44]. The PR3_R and PRPR3_MR models, to which machine learning was applied, presented similar statistics as that of FAO56 P–M with averages of 103.03 and 102.96 mm/month. For the error against FAO56 P–M, PR3_MR achieved the

lowest RMSE of 11.30 mm/month and MAE of 8.75 mm/month. The error of the Blaney–Criddle equation, which was widely used in the design of agricultural reservoirs in the past when meteorological data were not sufficient, was the highest at RMSE = 57.52 mm/month and MAE = 53.77 mm/month. In addition to the Thornthwaite equation, $R^2$ was also less than 0, which indicated no correlation at all. The Hargreaves equation showed a better error than the Blaney–Criddle and Thornthwaite equations; however, it achieved a larger error than the RMSE of 19.28 mm/month and MAE of 16.47 mm/month of PR3_R, which learned the monthly average temperature and diurnal temperature range. Thus, the machine learning method was observed to be more useful than the empirical equations when calculating the monthly $ET_0$.

**Table 6.** Comparison of the daily, monthly, and annual $ET_0$ from 2017 to 2021 for all sixty-two meteorological stations by $ET_0$ estimation methods.

| Models | Statistics | | | | Errors | | |
|---|---|---|---|---|---|---|---|
| | Max. | Min. | Avg. | Std. dev | RMSE | MAE | $R^2$ |
| *By daily parameters* | | | | | | | |
| FAO56 P–M | 9.80 | 0.76 | 3.38 | 1.30 | - | - | - |
| Hargreaves | 7.87 | 0.66 | 3.93 | 1.25 | 0.95 | 0.79 | 0.47 |
| PR3_R | 7.62 | 1.35 | 3.37 | 0.89 | 0.89 | 0.72 | 0.54 |
| PR3_RR | 6.73 | 1.05 | 3.37 | 1.03 | 0.72 | 0.56 | 0.69 |
| *By monthly parameters* | | | | | | | |
| FAO56 P–M | 171.55 | 42.25 | 103.28 | 25.89 | - | - | - |
| Hargreaves | 190.34 | 55.96 | 120.27 | 28.49 | 22.39 | 19.73 | 0.25 |
| Blaney–Criddle | 211.70 | 87.67 | 157.03 | 30.04 | 57.52 | 53.77 | −3.94 |
| Thornthwaite | 192.93 | 22.42 | 105.76 | 42.89 | 32.32 | 26.93 | −0.56 |
| PR3_R | 158.61 | 68.45 | 103.03 | 15.84 | 19.28 | 16.47 | 0.45 |
| PR3_MR | 149.31 | 50.74 | 102.96 | 21.43 | 11.30 | 8.75 | 0.81 |
| *By annual parameters* | | | | | | | |
| FAO56 P–M | 859.98 | 573.33 | 722.94 | 56.46 | | | |
| Hargreaves | 1016.38 | 643.33 | 841.91 | 71.67 | 149.00 | 134.99 | −5.96 |
| Blaney–Criddle | 1162.08 | 932.43 | 1099.24 | 36.88 | 378.75 | 376.30 | −44.0 |
| Thornthwaite | 831.50 | 570.26 | 740.34 | 43.02 | 44.44 | 37.20 | 0.38 |
| PR3_M | 792.02 | 614.77 | 723.55 | 32.54 | 41.32 | 32.94 | 0.46 |
| PR3_MR | 791.01 | 611.96 | 723.89 | 32.74 | 40.49 | 32.26 | 0.49 |

Max.: maximum, Min.: minimum, Avg.: average, Std.dev: standard deviation, RMSE: root-mean-square error, MAE: mean absolute error, and $R^2$: coefficient of determination.

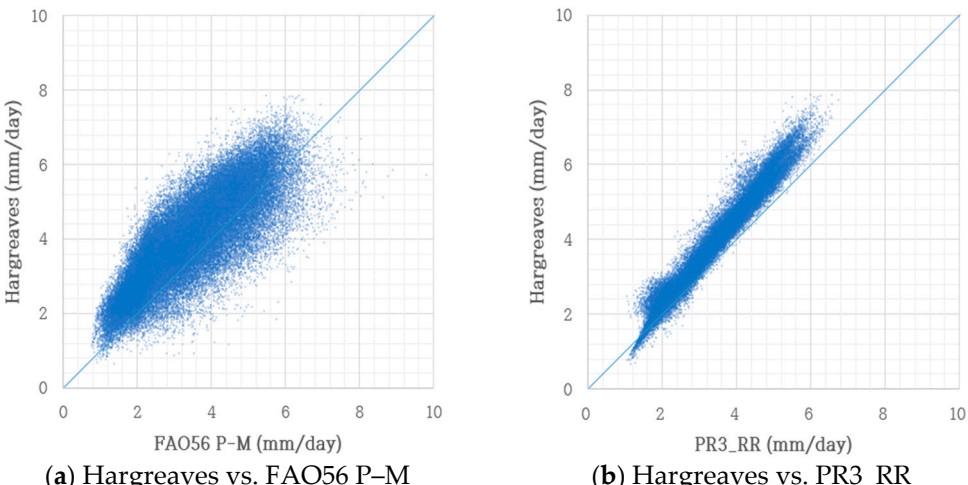

(**a**) Hargreaves vs. FAO56 P–M          (**b**) Hargreaves vs. PR3_RR

**Figure 8.** Daily $ET_0$ by Hargreaves compared with FAO56 P–M and PR3_RR.

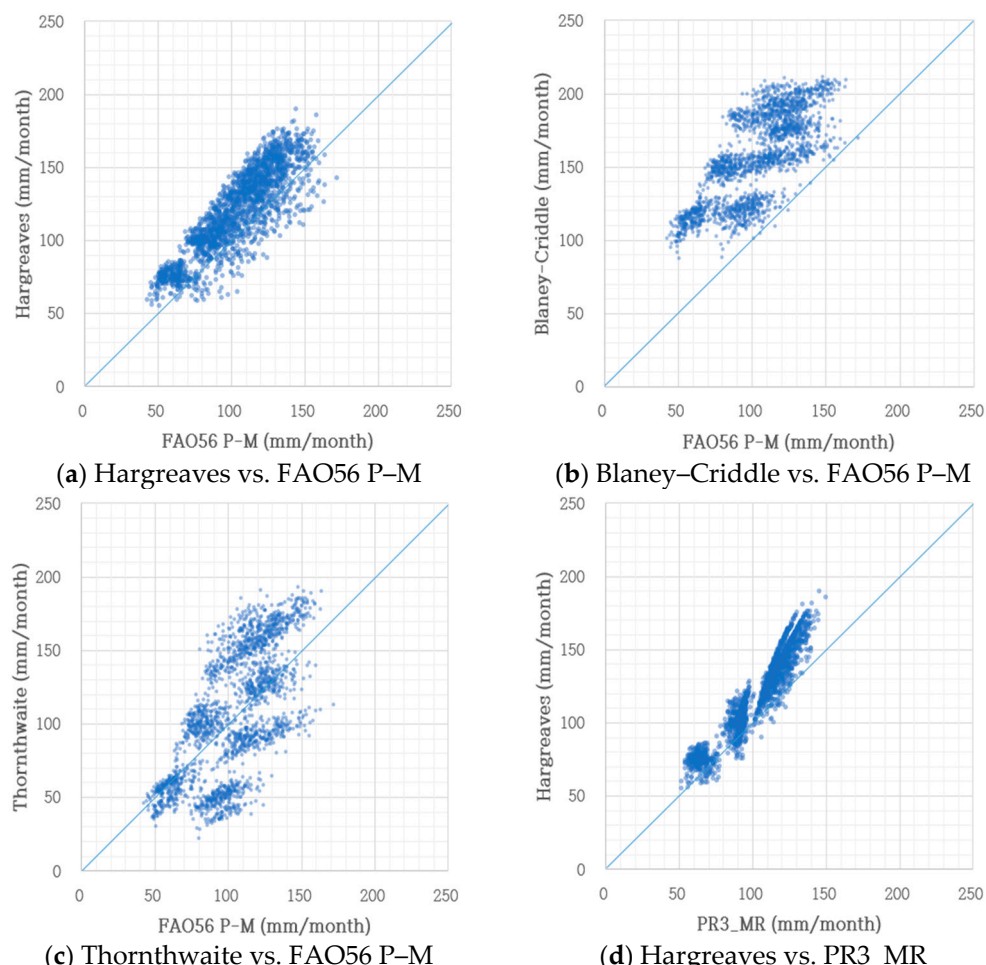

**Figure 9.** Monthly ET$_0$ by empirical equations compared with FAO56 P–M and PR3_MR.

For the annual evapotranspiration values, the PR machine learning method showed a lower error against FAO56 P–M than the empirical equations. The national average of annual ET$_0$ by FAO56 P–M was 722.94 mm/year, whereas the Hargreaves and Blaney–Criddle equations overestimated it to 841.91 and 1099.24 mm/year, respectively. For Thornthwaite, the average value was 740.34 mm/year and the maximum value was 831.50 mm/year, which was similar to the values for FAO56 P–M. The averages of PR3_MR and PR3_M, which showed the best performances in annual learning, were 723.89 and 723.55 mm/year, respectively; the maximum values were 791.01 and 792.02 mm/year, respectively, which suggest statistics closer to FAO56 P–M than those of the Hargreaves or Blaney–Criddle equations (Figure 10).

For the error with FAO56 P–M, the Hargreaves equation had a high error rate with an RMSE of 149.00 mm/year and $R^2$ less than 0. This meant that there was no correlation, and this was an abnormal case where the prediction performance was lower than the model predicted by the mean value. The RMSE of the Blaney–Criddle equation was excessive at 378.75 mm/year. However, the Thornthwaite equation had an RMSE of 44.44 mm/year with FAO56 P–M, which was the same for PR3_M and PR3_MR, with RMSEs of 41.32 and 40.49 mm/year, respectively. The Hargreaves or Blaney–Criddle equations were not appropriate for annual ET$_0$, and Thornthwaite showed the best error among the empirical equations. Similar to the daily and monthly results, the polynomial regression machine learning showed the closest results to the FAO56 P–M in annual reference evapotranspiration, and it was appropriate as an alternative to the temperature-based empirical equation.

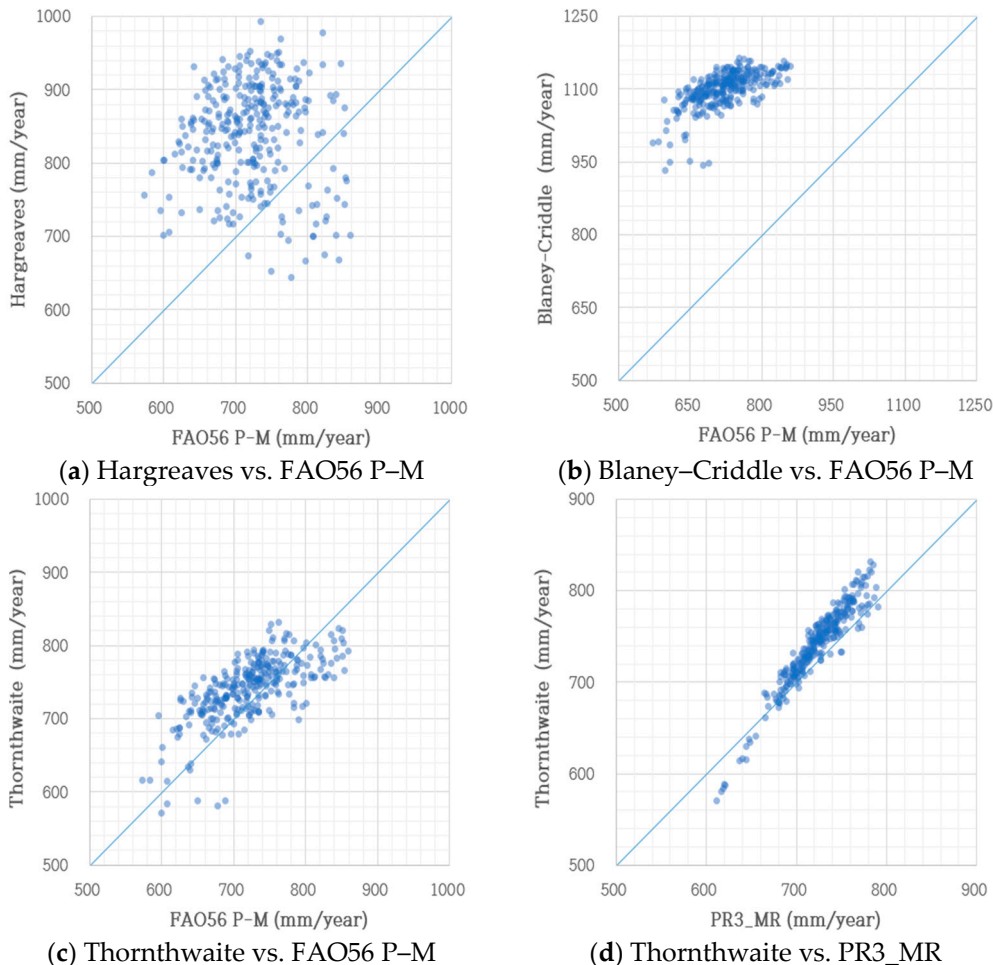

**Figure 10.** Annual $ET_0$ by empirical equations compared with FAO56 P–M and PR3_MR.

## 4. Conclusions

A linear regression machine learning model that can estimate the reference evapotranspiration under limited conditions of meteorological data was tested, and its reliability was reviewed by comparing it to FAO56 P–M. The applicability of the proposed method was evaluated by comparing it to temperature-based empirical equations, such as Hargreaves, Blaney–Criddle, and Thornthwaite. To this end, meteorological data obtained from sixty-two meteorological stations across the country collected over 41 years (1981 to 2021) were used by applying multiple linear regression (MLR) and polynomial regression (PR) algorithms using Python-based libraries, such as Scikit-Learn. A total of twelve learning models were tested based on the composition of input datasets, such as maximum temperature, minimum temperature, average temperature, diurnal temperature range, and extraterrestrial radiation. Using four performance indicators, such as MSE, RMSE, MAE, and $R^2$, the daily $ET_0$, monthly accumulated $ET_0$, and annual accumulated $ET_0$ by FAO56 P–M from 2017 to 2021, respectively, were used as controls, and the test performance of each learning model was evaluated on a daily, monthly, and annual scale.

The cubic PR machine learning model (PR3_MR or PR3_RR) showed the best performance in the daily $ET_0$, monthly $ET_0$, and annual $ET_0$ prediction tests. Compared to models that did not learn extraterrestrial radiation, models that learned the DTR or average temperature and extraterrestrial radiation together showed less errors in all tests. Furthermore, the PR algorithm showed a better performance than the MLR algorithm in all time scales (for example, the best $R^2$ were daily 0.62, 0.69; monthly 0.77, 0.81; annual 0.40, 0.49 MLR and PR algorithms, respectively) and predicted an $ET_0$ closer to FAO56 P–M than the temperature-based empirical equations. Therefore, if only the temperature data were

measured, they can be learned using the cubic PR machine learning model along with the extraterrestrial radiation data (calculated based on temperature-observation-point-location information). This proposed method can calculate the $ET_0$ of the smallest error compared to the FAO56 P–M, and can substitute the empirical equation.

This study proposed a linear regression learning model that has sufficient applicability compared to the empirical equations and can be applied to all of South Korea. However, inland and coastal areas are different, and there may be errors in predicting the $ET_0$ according to the spatiotemporal deviation of learning data. Thus, it is necessary to study the optimal model for each region by examining other training data and learning models considering regional characteristics.

**Author Contributions:** All authors contributed to the design and development of this manuscript. Conceptualization and methodology, M.-W.J. and S.-J.K.; data collection, S.-J.B.; implementation of machine learning algorithms, S.-J.K.; writing—original draft preparation, M.-W.J.; writing—review and editing, M.-W.J. and S.-J.B.; project administration, M.-W.J. All authors have read and agreed to the published version of the manuscript.

**Funding:** This research was funded by the National Research Foundation of Korea (NRF), grant funded by the Korea government (MSIT) (No. NRF-2019R1A2C1010125).

**Institutional Review Board Statement:** Not applicable.

**Informed Consent Statement:** Not applicable.

**Data Availability Statement:** Not applicable.

**Conflicts of Interest:** The authors declare no conflict of interest.

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
