# Peer review of "Linear Regression Machine Learning Algorithms for Estimating Reference Evapotranspiration Using Limited Climate Data"

_sustainability, doi:10.3390/su141811674_

Round 1

Reviewer 2 Report

The manuscript (ms) examines multiple linear regression and polynomial regression for ETo, employing ground-based data from sixty-two stations across South Korea, for 2017-2021. The regression models are compared against three common empirical methods with FAO PM as reference, on a daily, monthly and annual scale. The polynomial regression modela performed the best. The findings (errors) are satisfactory, especially on the annual scale. The study is useful in a parsimonious aspect of predicting ETo with limited data. The subject is updated and attracts scientific interest. However there are several weaknesses that should be addressed aiming the ms to be eligible for publication. 

I provide an attached pdf file to assist the process of revision. The main points are mentioned below:

The ms is poorly referenced and the same references are repeatedly cited. For exapmle, the first paragraph of the Introduction mentioning a wide range of subjects relevant to ETo is supported basically by a single reference. Moreover, the majority are not very recent (5 years or more), albeit there is a plethora of updated studies on ETo regression, in the very recent literature. (See the attached pdf file for sources).

Structure issue: The Results section plays the role of Results and Discussion. Results only need to present the findings. The elaboration/comparison etc must be carried out in the Discussion section which is missing here. Revise the Result section keeping Tabes, Figures and findings presentation wording and move all commenting in a Discussion section.

Lines 244-252 in Results: This passage should be moved in Methods, and be properly embedded in 2.3. (The software at the beginning of 2.3.) Cite the used software package.

Lines 261-265 in Methods: This comparison should be moved in Discussion.

Moreover: Why only those two references were selected to compare results with? They are not for the same area. It looks biased/ favorable since they report larger errors than yours. Consider to employ more (relevant) findings to compare with (see the links already provided to retrieve some) in a meaningful way (e.g. for the same area or the same inputs) and try to justify the causation of the differences to your outcomes, if possible.

Line 279 in 3.1: When (in)significance is mentioned, the corresponding p-value must be provided in parentheses.

Lines 300-301 and Figure 4 in 3.1 (also Lines 417-418). The negative values of R(Fig.4) need elaboration to support that they are not because of some mal-operation of the software you made or any inherent problems of the time-series.

Lines 349-351 in 3.3: Demands revision to convey the proper meaning (refer to the estimates). Because accumulating (actual values) is the ultimate method.

Figure 1 in not mentioned anywhere in text. Moreover, it should be moved after 2.1. Figure 1. is more informative than 2.1. paragraph (Linear-regression machine-learning models). This means that 2.1 should be enxpanded, explaining the steps displayed in Figure 1 in more detail.

Tables and Figures must be placed right after their first citation in text, not before as happends here.

There is moderate need English to be rechecked (mostly in terms of syntax and some specific words). Moreover, in Methods there are some very long sentences which need to be split for clarity (See the attached pdf file).

Line 267 (and elsewhere): Prefer "training and testing phases" (rather than "train and test"). Replace the word "existing" which is overly used in ms when referring to methods. It does not make sense.

The style of Tables need to be adjusted to journal's standards. The units have different fonts and need revision throughout the ms.

I recommend authors to address all the comments found in the pdf file.

Kind regards

Round 2

Reviewer 2 Report

My comments were satisfactorily addressed by the authors. The structure is now the proper, the references were updated, the justifications support all the assertions and the results are effectively conveyed. I recommend acceptance as is.

Kind regards